# NMDA Enhances and Glutamate Attenuates Synchrony of Spontaneous Phase-Locked Locus Coeruleus Network Rhythm in Newborn Rat Brain Slices

**DOI:** 10.3390/brainsci12050651

**Published:** 2022-05-16

**Authors:** Bijal Rawal, Vladimir Rancic, Klaus Ballanyi

**Affiliations:** Department of Physiology, Faculty of Medicine & Dentistry, University of Alberta, Edmonton, AB T6G 2H7, Canada; bijal@ualberta.ca (B.R.); rancic@ualberta.ca (V.R.)

**Keywords:** brain slices, glutamate, iGluR, local field potential, locus coeruleus, neuron, NMDA, noradrenaline, neonatal, oscillations, pattern transformation, rhythm generation, synchronization

## Abstract

Locus coeruleus (LC) neurons are controlled by glutamatergic inputs. Here, we studied in brain slices of neonatal rats NMDA and glutamate effects on phase-locked LC neuron spiking at ~1 Hz summating to ~0.2 s-lasting bell-shaped local field potential (LFP). NMDA: 10 μM accelerated LFP 1.7-fold, whereas 25 and 50 μM, respectively, increased its rate 3.2- and 4.6-fold while merging discrete events into 43 and 56% shorter oscillations. After 4–6 min, LFP oscillations stopped every 6 s for 1 s, resulting in ‘oscillation trains’. A dose of 32 μM depolarized neurons by 8.4 mV to cause 7.2-fold accelerated spiking at reduced jitter and enhanced synchrony with the LFP, as evident from cross-correlation. Glutamate: 25–50 μM made rhythm more irregular and the LFP pattern could transform into 2.7-fold longer-lasting multipeak discharge. In 100 μM, LFP amplitude and duration declined. In 25–50 μM, neurons depolarized by 5 mV to cause 3.7-fold acceleration of spiking that was less synchronized with LFP. Both agents: evoked ‘post-agonist depression’ of LFP that correlated with the amplitude and kinetics of V_m_ hyperpolarization. The findings show that accelerated spiking during NMDA and glutamate is associated with enhanced or attenuated LC synchrony, respectively, causing distinct LFP pattern transformations. Shaping of LC population discharge dynamics by ionotropic glutamate receptors potentially fine-tunes its influence on brain functions.

## 1. Introduction

The locus coeruleus (LC) in the dorsal pons is the source for noradrenergic innervation of most brain structures, and consequently controls multiple behaviors including arousal, sleep-wake cycle, breathing, memory, pain sensation, anxiety and opioid (withdrawal) effects [1,2,3]. In vivo, LC neurons show spontaneous pacemaker-like (‘tonic’) single action potential spiking or they discharge phasically during specific afferent synaptic inputs involving various neurotransmitters including glutamate [2,4,5,6,7]. The important functional role of glutamatergic innervation of the LC is evident from the fact that ionotropic glutamate receptor (iGluR) modulators notably affect brain functions [8,9]. For example, the NMDA-type iGluR antagonist ketamine has an acute antidepressant action that can reduce suicide ideation [10,11]. Moreover, iGluR activation is involved in transformation of tonic LC neuron spiking into oscillatory (‘burst’) discharge during morphine application to adult rats [12]. As stated in the latter report, dose–response studies are needed to find out whether both NMDA- and non-NMDA-type (i.e., AMPA/Kainate-type) iGluRs are involved in this opioid-evoked LC discharge pattern transformation. The authors also noted that it is not clear if their morphine injection primarily affected brain areas outside the LC. In that regard, particularly the nucleus paragigantocellularis, the lateral habenula and the prefrontal cortex send glutamatergic axons to the LC [4,5,13,14]. Electrophysiological recording in brain slices that are devoid of these and other afferent glutamatergic structures has established that activation of either type of iGluR accelerates tonic LC neuron spiking [15,16,17]. The lack of occurrence of NMDA and AMPA/Kainate receptor-dependent LC bursting in vitro might be due to the fact that responses of single neurons were monitored in the latter studies. Instead, local field potential (LFP) recording might have revealed a more complex LC network response as seen in various brain structures, both in vivo and in vitro [18,19,20,21]. In that regard, we found with suction electrode recording that the LC in newborn rat brain slices generates a spontaneous LFP at a rate of ~1 Hz [22]. As discussed in our more recent report, this LFP rhythm does not reflect phasic LC activity, but is rather the consequence of rhythmic subthreshold oscillations (STOs) involving intrinsic Ca^2+^-dependent ion conductances [23]. We reported in the latter study that the LFP pattern can transform to prolonged multipeak discharge during either recovery from opioid inhibition or raised extracellular K^+^. Such transformed LC discharge resembles NMDA-evoked rhythmic neuronal activities in slices of various brain circuits [24,25,26,27].

Here, we firstly used dose–response relationships to analyze whether iGluR activation by NMDA or glutamate mimics the K^+^- or opioid-evoked LFP pattern transformation. Subsequently, we combined LFP recording with membrane potential (V_m_) recording to analyze whether NMDA and glutamate effects on network rhythm are reflected by the above mentioned dose-dependent acceleration of single neuron spiking and at which dose their phase-lock might change.

## 2. Materials and Methods

### 2.1. Preparation and Solutions

The experiments were performed on horizontal brain slices from 0–5 days-old CD-001 (SD) rats of unknown sex (Charles River Laboratory Inc., Wilmington, MA, USA). All procedures were approved by the University of Alberta Animal Care and Use Committee, in compliance with the guidelines of the Canadian Council for Animal Care and in accordance with the Society for Neurosciences ‘Policies on the Use of Animals and Humans in Neuroscience Research’.

Procedures for generating LC-containing brain slices are described elsewhere in detail [22]. In brief, rats were anesthetized with 2–3% isoflurane to a level that caused disappearance of the paw withdrawal reflex. They were then decerebrated and the neuraxis was isolated at 18–20 °C in superfusate containing (in mM): 120 NaCl, 3 KCl, 1.2 CaCl_2_, 2 MgSO_4_, 26 NaHCO_3_, 1.25 NaH_2_PO_4_ and 10 D-glucose (pH adjusted to 7.4 by gassing with carbogen, i.e., a mixture of 95% O_2_ plus 5% CO_2_). The brain was glued on its ventral surface to a metal cutting plate, which was then inserted into the vise of a vibratome (Leica VT1000S; Leica Microsystems, Richmond Hill, ON, Canada). In carbogenated superfusate, horizontal brain slices were cut at room temperature, initially at 400–600 μm steps, until the 4th ventricle appeared. Once the LC appeared as a dark oval area close to the lateral border of the 4th ventricle, a single 400 μm thick slice was cut. This slice typically contained >50% of the dorsoventral aspect of the round LC soma area which extends in newborn rodents by ~300 μm in the horizontal plane [28].

For recording, a slice was mechanically fixed with a platinum ‘harp’ in an acrylic chamber (volume ~1 mL) with a glass bottom (Warner Instruments, Hamden, CT, USA). The LC and individual cells were visualized with a 20× objective (XLUMPlanF1, numerical aperture 1.0) of an MPE microscope (Olympus, Markham, ON, Canada) or an IR-DIC video camera (OLY-150, Olympus). A peristaltic pump (Sci-Q 403U/VM, Watson-Marlow, Wilmington, MA, USA) was used to apply, at a rate of 5 ml/min, carbogenated superfusate, which was removed from the chamber with a vacuum applied to a hypodermic needle. Superfusate temperature in the chamber was kept at 28 °C via a heat control system (Thermo-Haake DC10-V15/B, Sigma-Aldrich, Markham, ON, Canada).

### 2.2. Drug Application

Stock solutions for the agents (all from Sigma-Aldrich) were as follows: 25 mM (in H_2_O) for NMDA, 1 M (in H_2_O) for glutamate and 100 mM (in DMSO) for kynurenic acid, a non-competitive non-selective iGluR antagonist. The agents were applied via addition to the superfusate for two reasons. Firstly, our major aim was to investigate if (NMDA-type) iGluR activation is involved in LFP pattern transformation from ~0.3 s-lasting discharge to several seconds-lasting multi-peak events seen in our previous study during elevated K^+^ or recovery from opioid inhibition [23]. As 7 mM K^+^ and the µ-opioid receptor agonist DAMGO were bath-applied in that report, we used the same approach to study here whether this LFP pattern transformation is mimicked by bath-applied glutamate or NMDA. Secondly, bath application provides a steady-state extracellular concentration of the agents. This enabled determination of dose–response relationships for glutamate and NMDA to analyze (i) the extent to which a change in the LC population response reflects the above mentioned dose-dependent acceleration of spiking in single LC neurons [15,16,17] and (ii) at which dose phase-lock of spiking might change. A complete dose–response analysis could be done in individual slices by applying for 5 min each 5, 10, 25, 50 and 100 μM NMDA or 5, 10, 25, 50, 100, 250 and 500 μM glutamate. For testing effects on the LFP in control solution, 2.5 mM kynurenic acid was applied for 5 min. To analyze kynurenic acid effects during V_m_ recording, a LFP pattern-transforming glutamate dose was applied and kynurenic acid was then administered at 0.25–2.5 mM only until the onset of the blocking effect (typically after 1–2 min) to accelerate recovery during washout.

### 2.3. Electrophysiological Recording

Patch pipettes were pulled from borosilicate glass capillaries (GC-150TF-10; 1.5 mm outer Ø, 1.17 mm inner Ø, Harvard Apparatus, Holliston, USA) to an outer tip Ø of ~2 μm using a vertical puller (PC-10, Narishige International Inc., Amityville, NY, USA). They were used to record either (i) the LFP (after breaking and beveling the tip to obtain a suction electrode), (ii) extracellular spiking from a LC neuron pair or (iii) V_m_ in a single LC neuron. Electrophysiological signals were sampled at 4–20 kHz into a digital recorder (Powerlab 8/35 + LabChart 7 software, ADInstruments, Colorado Springs, CO, USA) connected to a personal computer.

Suction electrode recording: The major aim of this study was to analyze if NMDA or glutamate transform the LFP pattern. We reported [23] that LFP recording in the LC is difficult with fine-tipped microelectrodes, contrary to using suction electrodes often applied to monitor nerve activities or neuronal population rhythm within the isolated breathing center of newborn rodent brainstem slices [18]. For suction electrode LFP recording, patch pipettes were broken and subsequently beveled manually with sand paper (Ultra Fine 600 Grit, Norton-Saint Gobain, Worcester, WA, USA) at an angle of 45° to an oval-shaped tip with an inner opening of 40–60 μm. After filling with superfusate, the dc resistance of the suction electrodes was ~200 kΩ [23]. Following insertion into a patch electrode holder system (ESP-M15P and MHH-25, Warner Instruments), 15–30 mmHg suction was applied to the electrode with a syringe (BD Diagnostics, Franklin Lakes, NJ, USA) and controlled via a differential pressure sensor (Honeywell, SCX05DN, Fort Worth, TX, USA). A MP-285 micromanipulator (Sutter Instrument Company, Novato, CA, USA) was used to position the suction electrode opening at a flat angle of ~30° on the slice surface followed by application of <5 mmHg negative pressure. The ‘raw’ suction electrode signal was amplified (×10 k) and band-pass-filtered (0.3–3 kHz) using a Model-1700 differential amplifier (AM-Systems, Sequim, WA, USA). In parallel, the signal was processed for another recording channel by integration with a time constant of 50 ms using a MA-821/RSP unit (CWE, www.cwe-inc.com (accessed on 14 May 2022). This is a convenient way to represent neuronal rhythmic discharge as routinely used, e.g., for respiratory [18] and locomotor [29,30] network analyses.

Paired neuron spike recording: For extracellular single-cell spike recording, superfusate-filled unbroken patch electrodes with a dc resistance of 4–5 MΩ [22] were inserted into the above mentioned holder and two MP-285 micromanipulators (Sutter) were used to position them at an angle of ~30° on the slice surface close to a target cell. Repetitive single Na^+^ spike discharge was recorded in >90% of neurons when the electrode was pushed slightly against their soma and ~20 mmHg negative pressure was applied. Recordings were mostly stable for >5 min, and in ~50% of cases for >30 min. Spiking was recorded in current-clamp mode at 10× amplification using a patch-clamp amplifier (EPC-10 HEKA, Lambrecht, Germany).

Whole-cell recording: For neuronal V_m_ recording using an EPC-10 amplifier, patch pipettes were filled with (in mM): 140 K-gluconate, 1 NaCl, 0.5 CaCl_2_, 1 MgCl_2_, 1 Na_2_-ATP, 1 mM BAPTA and 10 Hepes (pH was adjusted to 7.4 with KOH; dc resistance in superfusate was 5–8 MΩ). When dimpling of the soma area occurred while visually targeting a neuron using the same manipulator as for extracellular recording, 20 mmHg positive pressure was released and negative pressure was applied for gigaseal formation (>1 GΩ). Whole-cell recording was established by abrupt suction (~100 mmHg). Series resistance, comprising access plus electrode resistance, was compensated during a test pulse at the beginning of a recording and was also checked, and eventually adjusted, later during the measurement. Access resistance typically ranged between 10–50 MΩ and was stable in >90% of neurons even during recordings lasting >1 h. For determining neuronal input resistance ranging from 120–370 MΩ, hyperpolarizing current pulses (50–100 pA) were injected, mostly at an interval of 10–15 s. Cells were analyzed only where spike amplitude was >70 mV while V_m_ was stable for a 5 min control period. Due to ongoing STOs [31,32], LC neurons do not have a ‘resting’ V_m_, which was thus defined as the value at 50% of the interval between spikes at the oscillation peak.

### 2.4. Data Analysis

LFP rates and amplitudes were quantified during 1 min recording time periods in control or at steady-state of drug effects. LFP duration was defined with Clampfit software (Molecular Devices Corporation, Chicago, IL, USA) as the time interval from when the averaged signal increased above and decreased below a threshold set at 10% of peak amplitude, respectively. The extent of LC network synchrony was determined with Clampfit software by comparing over a time period of 10 s the cross-correlation between whole-cell-recorded single neuron spiking and the integrated LFP. The peaks in cross-correlograms refer to the cross-correlation function estimate (CFE) values for the accuracy of synchrony. The lag time quantifies the shift in the peak between these events and thus gives a measure of spike jitter. This approach is applied to correlate a single neuron with (nerve) population discharge in respiratory [33] and locomotor networks [29]. For analysis of spike jitter, 20 LFP cycles and neuronal spiking were temporally aligned to the LFP peak using Clampfit software. Aligned traces were then overlapped and analyzed using a numerical matrix technique (Origin 6, Microcal Software). A summated LFP was obtained by averaging the 20 cycles. The average time point of neuronal spiking and its SD was used as a measure of jitter of their discharge.

The regularity of LC network discharge was determined by quantifying the irregularity (IR) score, which is an established parameter to analyze neural network rhythms, e.g., of the inspiratory center [34] using the following formula:

IR score = 100 * [Σ(Pn − Pn1)/Pn-1]/N, where N is the number of LFP events, Pn is the period of nth event and Pn-1 is the period of the preceding event. Note that the IR score has no unit. The lower the value, the more regular the network rhythm.

Values are given as means ± SD and n-values correspond to measurements in one slice per animal with a total of 58 rats used. Significance (non significant [ns]: *p* > 0.05, * *p* < 0.05, ** *p* < 0.01, *** *p* < 0.001) was assessed by a two-tailed paired t test and one-way ANOVA with Dunnett’s post-test (only applied for analysis of NMDA effects on LFP) using Prism software (GraphPad Software Inc., La Jolla, CA, USA).

To facilitate reading of the Results section text, quantified data shown with statistical details in the scatter plots of Figures 4, 6 and 7 are only represented as means ± SD.

## 3. Results

In a first series of experiments, LFP properties and dose-dependent LFP pattern changes were analyzed during NMDA or glutamate application. Subsequently, cellular responses were analyzed with combined LFP and V_m_ recording.

### 3.1. LFP Shape and Signal Analysis

We reported previously that neonatal rat LC neurons generate in a brain slice at ~1 Hz rhythmic ~0.2 s-lasting LFP events comprising non-synchronous instead of phase-locked mostly single spikes [23]. Here, we studied this population discharge in more detail to enable a comparison with potential LFP pattern transformations during NMDA or glutamate. The LFP was immediately seen when the suction electrode touched the slice surface in the LC neuron somata area. Its amplitude increased progressively until steady-state was reached after 0.5–1 h. The signal was stable, often for >5 h, particularly when <5 mmHg negative pressure was applied to the electrode. Using the same 40–60 μm outer tip Ø electrode, LFP amplitude and shape did not vary at different positions within the LC of a given slice.

In >70% of slices (n = 29), the rate, duration and amplitude of LFP were regular (Figure 1A–C). In five slices with regular rhythm and typical discharge shape, the LFP rate ranged from 0.80–1.40 Hz (mean 1.09 ± 0.22 Hz, i.e., 65 ± 14 events/min). LFP duration ranged from 0.13 to 0.2 s (mean 0.16 ± 0.02 s) and integrated LFP amplitude was 0.65–0.97 mV (mean 0.82 ± 0.13 mV). In the other < 30% of slices, the rhythm was more irregular and LFP events could show several peaks evident in the integrated signal (Figure 1D). Figure 1E shows the overlayed traces of five consecutive LFP events from each of the five slices. The mean of the integrated trace was fitted well with a bell-shaped Gauss function (R^2^ value: 0.98) (Figure 1E).

The LFP could show an irregularity regarding rate, duration and amplitude not only in distinct slices, but also between cycles in the same slice (Figure 1). This is due to the fact that not every neuron spikes during each cycle and phase-lock of their spiking, related to that of other neurons and the LFP, shows a jitter [23]. This was also seen here in the LFP plus extracellular spike recording from five neuron pairs (Figure 2). The suction electrode was positioned at the rim of the LC close to the 4th ventricle whereas the spike-detecting patch electrodes were positioned in randomly chosen spots within the LC (Figure 2A). There was no correlation between spike pattern and patch electrode position. Due to the variability of spiking that overlaps to form the LFP, it was not possible to determine exact numbers of contributing neurons. However, each LC neuron typically discharges only one action potential per cycle or shows random spike failures (Figure 2B,C). Thus, based on an estimate of the number of partially overlapping spikes in the raw signal, the LFP typically represents summated discharge in 3–10 cells. In some cases, the LFP contained large amplitude spikes from 1–3 neurons whose somata were likely located closer to the electrode than cells with smaller amplitude signals (Figure 1A–D).

### 3.2. NMDA- and Glutamate-Evoked LFP Pattern Transformation

The above findings indicate that the LFP envelope is bell-shaped due to a normal distribution of the occurrence of single spikes that jitter similarly around the LFP peak. Next, dose–response relationships were studied for the effects of NMDA and glutamate to determine the thresholds for putative LFP pattern transformation and elucidate whether high concentrations of the agents perturb network rhythm or cellular spiking as previously reported [15].

NMDA effects on LFP: Examples are given in Figure 3 while statistical analysis from very similar findings in four slices is shown in Figure 4. At 10 μM, 5 min NMDA application slightly increased the LFP rate and shortened its duration (Figure 3A). At 25–50 μM, NMDA enhanced these effects during the first 1–4 min of application when separate events merged to sinusoidal, still bell-shaped oscillations with <15% amplitude fluctuations due to tonic activity, indicating less phase-locked spiking (Figure 3B–E). During the next 1–2 min, LFP oscillations were rhythmically depressed for ~1 s (Figure 3B–D). During the resulting LFP ‘oscillation trains’, the rate of oscillations further increased slightly, but single event duration did not decrease more. After 2–5 stable cycles, LFP oscillation trains shortened in durations with declined oscillation amplitude and regularity toward their end (Figure 3C,D,F). The onset of oscillation trains and their progressive diminution was less delayed at 50 and 100 μM NMDA than at 25 μM, transforming rhythm later during the application into ‘spikes’ before the LFP was blocked (in three cases at 100 μM and one case at the end of 50 μM) (Figure 3F). The modest stimulatory NMDA effects at 10 μM, and initially at 25 μM, were consistent for the four slices. Contrarily, transition of faster LFP events into oscillation trains occurred at 25 μM in two of these slices and at 50 μM in the other two cases.

In the four slices, LFP duration decreased from 183 ± 38 to 133 ± 18 ms at 10 μM, 105 ± 19 early at 25 μM, 78 ± 23 ‘late’ (i.e., during oscillation trains) at 25 μM and 81 ± 5 ‘early’ at 50 μM (Figure 4A,B). Conversely, the rate of the still bell-shaped LFP (see inset in Figure 4) increased (in events/min) from 77 ± 25 in control to 126 ± 19 (1.7-fold) at 10 μM, 250 ± 30 (3.2-fold) ‘early’ at 25 μM, 356 ± 46 (4.6-fold) ‘late’ at 25 μM and 347 ± 48 (4.5-fold) ‘early’ at 50 μM (Figure 4A,C). LFP amplitude did not change at any NMDA dose (Figure 4A,D).

Amplitude fluctuations of early LFP oscillations tended to increase from <15% at 25 μM to 40–50% at 50 μM. When LFP oscillation trains at 50 μM (n = 4) were stable in their early phase for 3 cycles, their duration was 6.06 ± 2.64 s and rhythmic LFP blockade lasted 0.94 ± 0.34 s.

Glutamate-effects on LFP: An example is given in Figure 5 for one of the six slices tested. In four slices at 25 μM, and the other two at 50 μM, glutamate made the rhythm more irregular with appearance of variable amplitude and duration events that included in four cases crescendo-shaped multipeak discharge (Figure 5A,B). Applying glutamate in these slices at one dose above the pattern-transforming one (i.e., at 50 or 100 μM) firstly had the same effect. After 1–2 min though, profound tonic discharge developed and activity decreased 1–2 min later (Figure 5C). In total, 100, 250 and 500 μM glutamate caused initially similar tonic discharge, which then progressively declined until rhythm stopped at 250 μM in four of the six slices and in one case each at 100 or 500 μM (Figure 5D). In two slices, irregular small amplitude ~0.05 s-lasting LFP spikes persisted during a blockade of rhythm at 250 μM (not shown). The effects could only be quantified in the four slices (two in 25 and two in 50 μM) that showed separate crescendo-shaped multipeak discharge. In these cases, LFP duration increased from 232 ± 27.7 ms in control to 624 ± 150 ms (*p* = 0.002) (i.e., 2.7-fold), whereas the LFP rate showed a trend to rise from 42.3 ± 7.2/min to 57.3 ± 15.6 events/min (*p* = 0.13).

### 3.3. NMDA and Glutamate Effects on Network Synchrony and Irregularity

The above described NMDA-evoked shortening of LFP event duration indicates an increase in network synchrony, whereas increased LFP duration and irregularity indicates the opposite for glutamate. To substantiate this assumption, cross-correlation analysis was performed between the LFP peak and intracellular spike occurrence.

Glutamate-evoked attenuated synchrony: Cross-correlation analysis of glutamate effects was done in six slices. In three cases, 25 μM transformed the LFP pattern within 2–3 min. In the other three cases, the glutamate dose was increased after 5 min, from 25 μM to 50 μM, which transformed the pattern after a further 2–3 min. An example for the latter approach is shown in Figure 6A. LFP pattern transformation by 50 μM glutamate was accompanied by a 7 mV depolarization of whole-cell-recorded V_m_ from its ‘resting’ value of −45 mV. This depolarization accelerated spontaneous discharge in this neuron from 48 to 228 spikes/min. The correlogram for the recording in Figure 6A showed a shallower peak in glutamate vs. control, indicating decreased synchrony, and, correspondingly, the CFE value decreased from 0.5 to 0.35, whereas time lag increased from 90 to 150 ms (Figure 6B). The scatter plots for five neurons show that the effects were significant (Figure 6C). Further spiking in one apparently similar responding neuron did not cross-correlate with LFP and thus could not be analyzed. For the six neurons, mean values were −47.3 ± 2.9 mV for resting V_m_, 5.0 ± 1.1 mV (*p* < 0.001) for the glutamate depolarization and 52.1 ± 4.6 vs. 194 ± 61.0 spikes/min (*p* < 0.001) (i.e., a 3.7-fold acceleration) for control vs. glutamate.

NMDA-evoked enhanced synchrony: Based on the above results from the dose–response relationships, 32 μM NMDA was applied to seven neurons for cross-correlation analysis, as this dose should rapidly transform the LFP pattern into oscillations while not immediately evoking oscillation trains. As exemplified in Figure 7A, 32 μM NMDA depolarized a neuron early during LFP pattern transformation by 10 mV from −50 mV, and spike rate increased from 39 to 336 spikes/min. The correlogram for this recording revealed enhanced synchronization in NMDA evident as a CFE value increase from 0.32 to 0.73 and a concomitant decrease of time lag from 110 to 3 ms (Figure 7B). The scatter plots for five neurons show that the effects were significant (Figure 7C). Spiking in the other two similarly responding neurons did not cross-correlate with LFP and could thus not be analyzed. For the seven neurons, mean values were −48.6 ± 2.4 mV for resting V_m_, 8.4 ± 2.3 mV (*p* < 0.001) for the NMDA depolarization and 49.4 ± 6.1 vs. 355 ± 35.6 spikes/min (*p* < 0.001) (i.e., a 7.2-fold acceleration) for control vs. NMDA.

In line with the assumption based on these findings that network synchrony was enhanced, the jitter of the time point of spiking in these neurons related to the LFP decreased from 103.8 ± 34.4 ms in control to 13.9 ± 2.25 ms in NMDA (i.e., to 15.2 ± 7.2% of control) (Figure 7D).

Regularity of rhythm: A further measure of rhythmic neural network behavior is the irregularity score [34]. For the five slices used for the cross-correlation analysis, the irregularity score decreased from 54.6 ± 12.4 in control to 9.8 ± 1.7 in NMDA, indicating that the agent makes network rhythm more regular (Figure 7E). Regarding glutamate, for the five slices used for the cross-correlation analysis, the irregularity score did not change (50.7 ± 8.3 in control vs. 51.3 ± 14.0 in glutamate), indicating that the agent has no effect on the regularity of LC network rhythm.

### 3.4. V_m_ Changes during NMDA-Evoked Oscillation Trains

Next, it was studied how NMDA-evoked LFP oscillation trains correlate with the changes of neuronal V_m_. A total of 32 μM NMDA was used again, as oscillation trains in 25 μM tended to start only after several minutes, whereas 50 μM depressed them too quickly. Figure 8 exemplifies this for the neuron of Figure 7A,B,D. During rhythmic LFP inhibition starting 4 min into NMDA, V_m_ repolarized for ~0.5 s close to its resting value of −48 mV with concomitant spiking arrest. Then, a crescendo-like ‘ramp’ depolarization developed that caused spike onset after ~0.5 s, when LFP oscillations also recurred (Figure 8A). Between the onset and the end of the ramp depolarization by 18 mV, spike rate increased from a value slightly lower than at steady-state before start of the trains to a value slightly higher with a corresponding amplitude decline. Within the next 30 s, LFP oscillation trains and ramp depolarizations shortened, the latter with a steeper slope and more depolarized peak (between −28 and −30 mV). Spike rate changed similarly but showed a more pronounced amplitude decrease (Figure 8B).

To test if rhythmic depolarization is network-dependent, V_m_ of this neuron was hyperpolarized 30 s later to −68 mV by injection of −100 pA dc current (Figure 8B). Rhythmic depolarization persisted (as also in other 3 cells tested), but at a shorter duration with overlaying repetitive spiking, along with LFP transformation into small amplitude decrescendo-shaped events. At 6 min into NMDA, the LFP was blocked, whereas some random spiking persisted and the neuron was depolarized further, now showing steep offset rhythmic hyperpolarizations with an initial ‘rebound’ spike. All effects reversed 12 min after start of washout.

For six neurons, several parameters were analyzed for the most stable and longest oscillation train early during their appearance. Specifically, the train lasted for 7.1 ± 0.9 s, NMDA caused at the end of the train a 20.0 ± 6.3 mV depolarization from a resting V_m_ of −48.3 ± 2.5 mV (*p* = 0.0006) and spike rate during the last second was 462 ± 45.7 spikes/min compared to 49.3 ± 6.7 spikes/min in control (*p* < 0.001) (thus 9.3-fold faster). V_m_ repolarized after the end of the train to −46.8 ± 2.9 mV (*p* = 0.06) and the pause until the next train started was 1.0 ± 0.1 s.

### 3.5. Post-Agonist Depression (PAD)

Apart from their opposite action on LC neuron synchrony, both agents exerted a similar inhibitory effect. Specifically, within 0.5–2 min following the start of washout of NMDA (Figure 9A) or glutamate (Figure 9B), V_m_ hyperpolarized, and, at the same time, cellular spiking and LFP were blocked.

In four neurons, NMDA had depolarized V_m_ at the start of such PAD from a resting value −48.7 ± 2.5 mV by 22.5 ± 6.4 mV (*p* = 0.006). V_m_ hyperpolarized during the inhibition maximally to −65.7 ± 5.0 mV (*p* = 0.005) and the effect was reversed after 9.0 ± 1.4 min. In four different neurons, glutamate had depolarized V_m_ at the start of PAD from −48.5 ± 3 mV by 5.2 ± 3.2 mV (*p* = 0.04), the maximal hyperpolarization was to −52.0 ± 1.6 mV (*p* = 0.03) and the effect was reversed after 1.5 ± 0.4 min.

### 3.6. Receptor Specificity of Glutamate Effects

NMDA binds selectively to NMDA-type iGluR, whereas glutamate can act on NMDA- and AMPA/Kainate-type iGluR [35] plus diverse types of metabotropic glutamate receptors [36]. To study whether metabotropic receptors contribute to the observed glutamate effects, the non-competitive unselective iGluR antagonist kynurenic acid was used.

In the neuron of Figure 10A, kynurenic acid (500 μM) was applied during LFP pattern transformation due to 50 μM glutamate. Within 20 s, LFP was blocked along with a 15 mV hyperpolarization and spike arrest. Several seconds later, V_m_ started to repolarize. Intracellular spikes and LFP recurred after 80 s and control-like LFPs and spike rates were seen in the presence of glutamate and kynurenic acid after 3 min (Figure 10A). Subsequent washout of kynurenic acid in glutamate re-established the glutamate effects within 3–5 min (not shown). In four neurons, glutamate had depolarized V_m_ at the start of kynurenic acid-evoked hyperpolarization from a resting V_m_ of −44.7 ± 0.5 mV by 3.5 ± 0.5 mV (*p* = 0.001), the maximal hyperpolarization was to −52.5 ± 2.8 mV (*p* = 0.009) and the effect reversed after 2.3 ± 0.4 min.

The late countering kynurenic acid effect shows that glutamate effects are mediated by iGluR. To exclude a novel type of inhibitory effect of kynurenic acid in the initial phase of application, 2.5 mM of the agent was added to control superfusate in 4 slices with regular LFP pattern. Kynurenic acid had no effect (Figure 10B) as in our previous study [37].

### 3.7. Persistence of Single Neuron Spiking during Network Inhibition

NMDA- and glutamate-related blockade of intracellular spiking correlated well with inactivation of the majority of the 3–10 neurons comprising the LFP. However, early during kynurenic acid application in glutamate small amplitude random LFP spiking was seen, whereas normal LFP events and neuronal spikes were blocked (Figure 10A). Similarly, random activity of some neurons persisted during inhibition between NMDA-evoked oscillation trains, blockade of rhythm by the agents at high doses and PAD (Figure 3D, Figure 8B and Figure 9A,B). This indicates that some neurons are resistant to NMDA- and glutamate-evoked inactivation or PAD.

## 4. Discussion

We found that NMDA accelerates the LFP rhythm and enhances the synchrony of LC neuron spiking, whereas glutamate makes the rhythm less regular and weakens synchrony. In a *Perspective* article in this Special Issue [32], we refer to the current view that this spontaneous rhythm is due to intrinsic Ca^2+^-dependent ion conductances causing rhythmic depolarization of gap junction-coupled neonatal LC neurons. However, as also noted in the latter report, phasic afferent activity in the slice model can also be studied, namely with tetanic stimulation of slice areas outside the LC soma region. The present study is the first demonstration of LC neural network discharge pattern transformation by iGluR. We also found that NMDA and glutamate evoke PAD. Possible mechanisms and consequences for LC network organization are discussed in the following.

### 4.1. LFP Recording of LC Discharge Pattern Transformation

In our previous study, we used suction electrode recording to unravel that LC neurons show jittered spiking that is yet phase-locked to generate rhythmic ~0.2-lasting LFP discharge [23]. Here, we found that the integrated LFP has a bell-shaped signal envelope during both control and NMDA stimulation. This indicates that the jitter of spiking in an estimated 3–10 neurons comprising this signal distributes equally around a center of activity representing the LFP peak. In the latter report, we also showed that the LFP pattern changes to ~3 s-lasting multipeak events by either increasing LC excitability with raised extracellular K^+^ or during recovery from opioid inhibition [23]. Here, glutamate and NMDA elicited different LFP pattern transformations indicating that neither K^+^ nor opioids change LC excitability to cause release of glutamate, which might then act on oscillation-generating NMDA receptors (for details, see below). Glutamate elicited LFP pattern transformation at 25–50 μM. This can be functionally relevant in the intact neonatal LC as extracellular glutamate rises in vivo from a 1–10 μM baseline to >20 μM like during treatment with μ-opioid (ant)agonists [5,38,39].

The finding that kynurenic acid reversed the glutamate-evoked LFP pattern transformation shows lack of involvement of metabotropic glutamate receptors. Contrary to our observation of distinct LFP network discharge pattern transformations, single neuron recording here and previously [15,16,17,40] showed that glutamate and iGluR agonists only accelerate tonic spiking. As evident from this and the following discussion, LFP recording is a powerful tool to study the clearly complex neonatal LC network.

### 4.2. LC Synchrony Analysis with Combined LFP and Single Neuron Recording

NMDA-evoked LFP oscillations were faster, shorter and more regular than control events, whereas glutamate made the rhythm apparently more irregular and more than doubled the duration of separate events if present. The assumption that NMDA enhances and glutamate and weakens neonatal LC synchrony was confirmed by cross-correlating neuronal spiking with LFP events (Figure 6 and Figure 7). Moreover, NMDA notably reduced intracellular spike jitter. Regarding mechanisms for these transformations, neonatal LC rhythm generation does not involve ‘classical’ neurotransmission because the LFP persists during iGluR blockade, as shown by us with kynurenic acid [37] and GABA_A_ plus glycine receptor blockades [32], similar to persistence of single LC neuron spiking in slices [40,41,42]. The latter results and the inhibitory effect of gap junction blockade on spike synchrony established that neonatal LC neurons are connected via electrical synapses [28,31,42,43,44,45,46]. Consequently, NMDA and glutamate may enhance and weaken electrical coupling, respectively. As also discussed in our *Perspective* article in this issue [32], NMDA stabilizes synchronized LFP oscillations in gap junction-coupled inferior olive neurons of rat slices involving Ca^2+^/calmodulin-dependent protein-kinase-I activation, which enhances weak coupling of non-neighboring neurons [47]. Contrarily, in the adult rat LC neuromodulators presumably do not directly counteract the postnatal decrease of gap junction coupling [45] and instead neuromodulator-evoked spike slowing itself reverses this decrease [42]. Whereas the latter study did not reveal a correlation in neonatal rats between spike rate and gap junction-coupling, we found that faster spiking increases (NMDA) or decreases (glutamate) LC network synchrony. We hypothesize that incomplete neonatal LC synchrony reflects neuronal differences regarding spontaneous neuromodulator release, which changes V_m_ close to STO peaks, thus altering spike threshold [23]. Thus, glutamate-evoked desynchronization may also occur if LC neuron subclasses express different types and/or numbers of AMPA/Kainate receptors and are consequently depolarized to a different extent, thus reaching spike threshold at diverging times. This is discussed in detail below.

### 4.3. PAD and Periodic NMDA-Evoked Inhibition

Whereas LFP pattern transformation was associated with increased spiking, NMDA and glutamate also acted in an inhibitory way. Specifically, early during washout of both agents, neuronal hyperpolarization was associated with PAD of LFP, whereas periodic inhibition of LFP oscillations developed later during NMDA application. In vivo, PAD occurs after sensory stimulation-evoked discharge [48,49]. Regarding the mechanism, it was shown in adult rat slices that an AMPA/Kainate receptor-triggered Na^+^-dependent K^+^ current causes PAD after glutamate, but not NMDA [17]. As both agents were effective here, we rather hypothesize that SK-type Ca^2+^-activated K^+^ channels are involved as ongoing neuronal depolarization likely leads to a cytosolic Ca^2+^ rise mainly mediated by voltage-activated Ca^2+^ channels [50]. In fact, in adult rat LC neurons in vivo [51] and slices [52], current-evoked spike numbers correlate with the duration of a pronounced hyperpolarization during PAD that is attenuated by experimentally increased cellular Ca^2+^ buffering. Moreover, the pacemaker rate in LC neurons of mouse slices is regulated by cooperation of L- and T-type Ca^2+^ with SK2 channels [53,54]. Enhanced cooperation of these channels may also contribute to the NMDA-related periodic interruption of LFP oscillations. As an explanation, the intracellular correlate of oscillation trains is a rhythmic crescendo-like depolarization and concomitant progressively enhanced spiking followed by hyperpolarization-related PAD possibly starting when Ca^2+^ rises reach SK channel activation threshold. Our finding that NMDA-evoked neuronal rhythmic depolarization and spiking persists during current-evoked hyperpolarization indicates that these events are voltage-clamped by gap junction-coupled neighboring neurons as in the juvenile rat CA3 hippocampus [24]. Pharmacological analysis beyond the scope of this study is required to analyze underlying cellular mechanisms that differ notably between neuron types showing similar NMDA-evoked V_m_ oscillations [24,25,26,27].

### 4.4. Potential Role of Modular LC Organization on LFP Pattern Transformation

We hypothesized above that opposite iGluR effects on LFP patterns may be caused by differences in neuronal properties. Indeed, ventrally-located adult rat LC neurons with shorter spikes and smaller afterhyperpolarizations than LC core neurons act as a ‘pontospinal-projecting module’ [55], whereas dorsomedially-located small and densely packed GABAergic neurons in juvenile mice show faster spiking with enhanced adaptation [56]. Topographically distinct LC modules also exist regarding afferent synaptic inputs and efferent projections [3,7,57]. In neonatal rats, intrinsic pacemaker and burst properties and differences in synaptic responses seem to exist between subgroups of LC neurons, as noted in our *Perspective* article in this Special Issue [32]. Two findings here support the concept of functional LC modules in newborn rats: (i) 3 of 13 neurons were uncoupled from network bursting as revealed with cross-correlation analysis and (ii) spiking of some LFP neurons persisted during oscillation train inhibition, blockade of rhythm at high doses of NMDA or glutamate and (kynurenic acid-evoked) PAD (Figure 3D, Figure 8, Figure 9 and Figure 10A). If some LC neurons are more sensitive to iGluR activation than others, these cells may release noradrenaline within their module to change the activities of neighboring neurons and astrocytes. For example, if astrocytes are then activated via their α_1_ receptor, they may release lactate to excite in positive feedback LC neurons via a novel receptor [58]. Our bath-application approach for NMDA and glutamate was adequate for the aim of the present study. However, in future work the agents should also be applied focally to study whether LC modules differ in their sensitivity to a specific iGluR agonist or different iGluR subtypes. Additional experimental strategies are discussed in our *Perspective* article in this issue [32].

Our finding that already complex LC population discharge can transform into different LFP patterns during NMDA and glutamate (here) or during high K^+^ and opioids [23] indicates that neuron-astrocyte modules cooperate within the LC. There seems to be a causal link between pattern transformations, opioids and iGluR. Specifically, transformation of tonic spiking into oscillatory (‘burst’) discharge during μ-opioid receptor activation in adult rat LC neurons in vivo was reversed by iGluR blockade with kynurenic acid [12]. The authors hypothesized in this study that iGluR-dependent LC neuron bursting is important for opioid tolerance and dependence. In fact, modulation of (LC) neuron iGluR activity has a plethora of behavioral effects [8,9,10,11] that might critically depend on LC neuron discharge pattern. Similarly, pulse trains (but not single pulses) applied to the adult rat LC in vivo elicit changes in medial prefrontal cortex activity resembling those in LC noradrenaline input-dependent memory tasks [59]. A related study from the latter group proposed that the temporal structure of [LC-mediated] noradrenergic modulation may dynamically enhance or attenuate cortical responses to stimuli [60]. Moreover, multi-unit recordings in the adult rat LC in vivo revealed only synchrony in few neuron assemblies while they discovered novel infra-slow (0.01–1 Hz) fluctuations of LC unit spiking [21]. These in vivo findings and related recent results [3,20,57,61,62,63] based on LFP and/or multi-unit recordings established the current view that the adult LC is a complex and differentiated neuromodulatory system. Our in vitro findings here extend this view to the neonatal LC and future studies will likely unravel novel neuropharmacological interactions in its neuron-astrocyte modules. A potent tool for comparing discrete electrical activity patterns of multiple LC neurons with those in vivo is to study slices positioned on a multi-microelectrode array [64].

## Figures and Tables

**Figure 1 brainsci-12-00651-f001:**
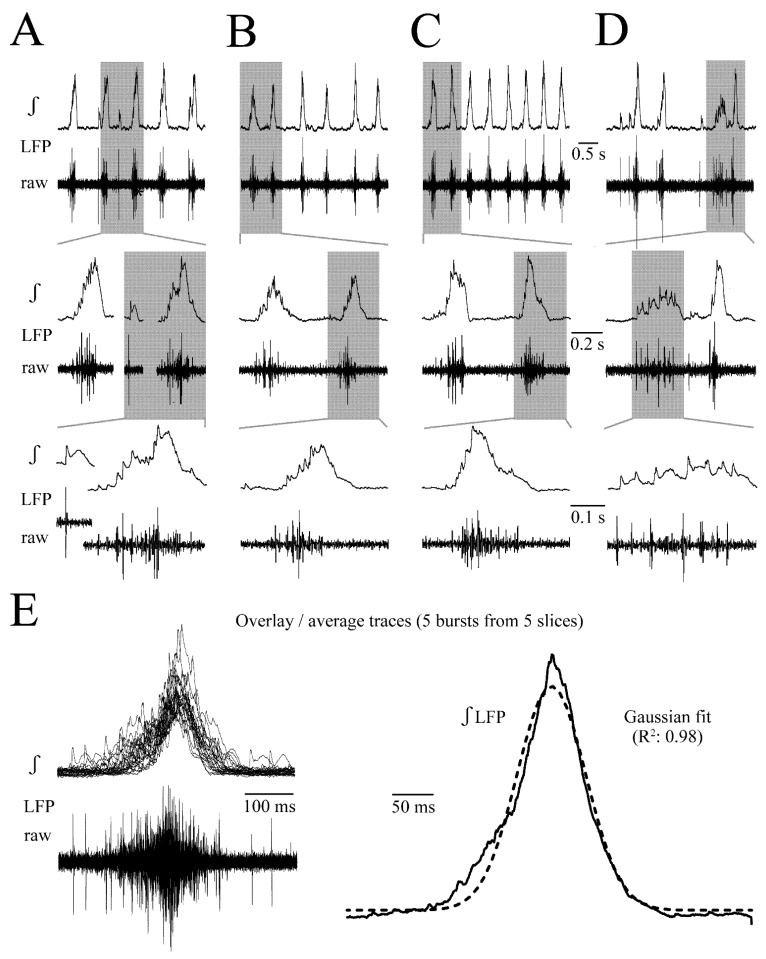
**Spontaneous local field potential (LFP) in newborn rat locus coeruleus (LC) slices.** (**A**) LFP was recorded with 40–60 μm outer tip Ø superfusate-filled suction electrodes positioned on the slice surface in the LC neuron somata area [see (Figure 2A)]. The differentially-amplified (×10 k) and bandpass-filtered (0.3–3 kHz) ‘raw’ signal was also recorded after integration (ʃ, time-constant 50 ms). LFP examples in (**A**–**C**) from three different slices show that the integrated signal has a similar shape and occurs at a quite regular rate and amplitude. However, displaying individual LFP events at higher time resolutions in the middle and lower traces indicates in raw traces that active neuron numbers, their distance from the electrode (indicated by spike amplitude) and time points of their discharge can differ. Additionally, in ~30% of slices, the LFP pattern is less regular and smaller amplitude events typically show a more dispersed pattern (**D**). (**E**) Traces on the left overlaid from five consecutive LFPs of the slices shown in (**A**–**C**), plus two further slices with similarly regular patterns. Traces on the right represent the average of integrated traces on the left plus a Gaussian fit for its kinetics revealing a bell-shaped LFP envelope.

**Figure 2 brainsci-12-00651-f002:**
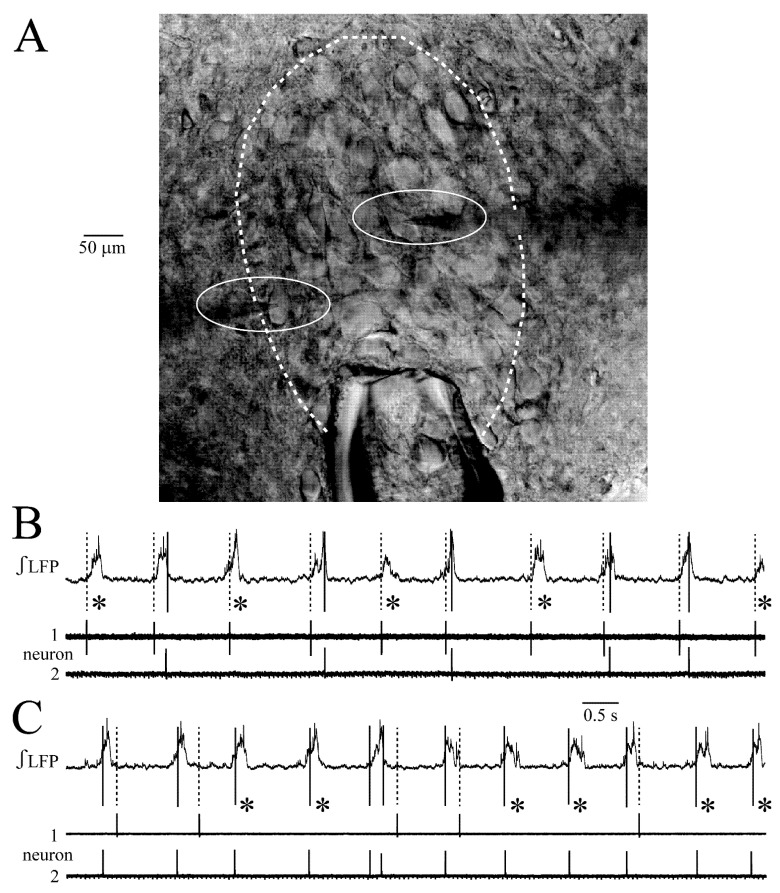
**LC neuron spike jitter.** (**A**) Photograph of the LC (outlined by white dashed line) with the suction electrode positioned close to the rim of the neuron somata area next to the 4th ventricle (not shown) while two patch electrodes (outlined by solid white lines) were positioned on the somata of LC neurons for spike recording. (**B**,**C**) Examples for such recording in two slices. In both examples, with (**B**) corresponding to the photo in (**A**), one neuron fires a single spike during each LFP whereas the other cell shows spike failures (see asterisks). The LFP comprises mostly discharge from 3–10 neurons (see Results). Thus, random spike failures cause differences in LFP amplitude, whereas jitter regarding the time point of discharge during the LFP [see distance between solid and dotted lines and compare (Figure 7D)] explains LFP duration and shape variability.

**Figure 3 brainsci-12-00651-f003:**
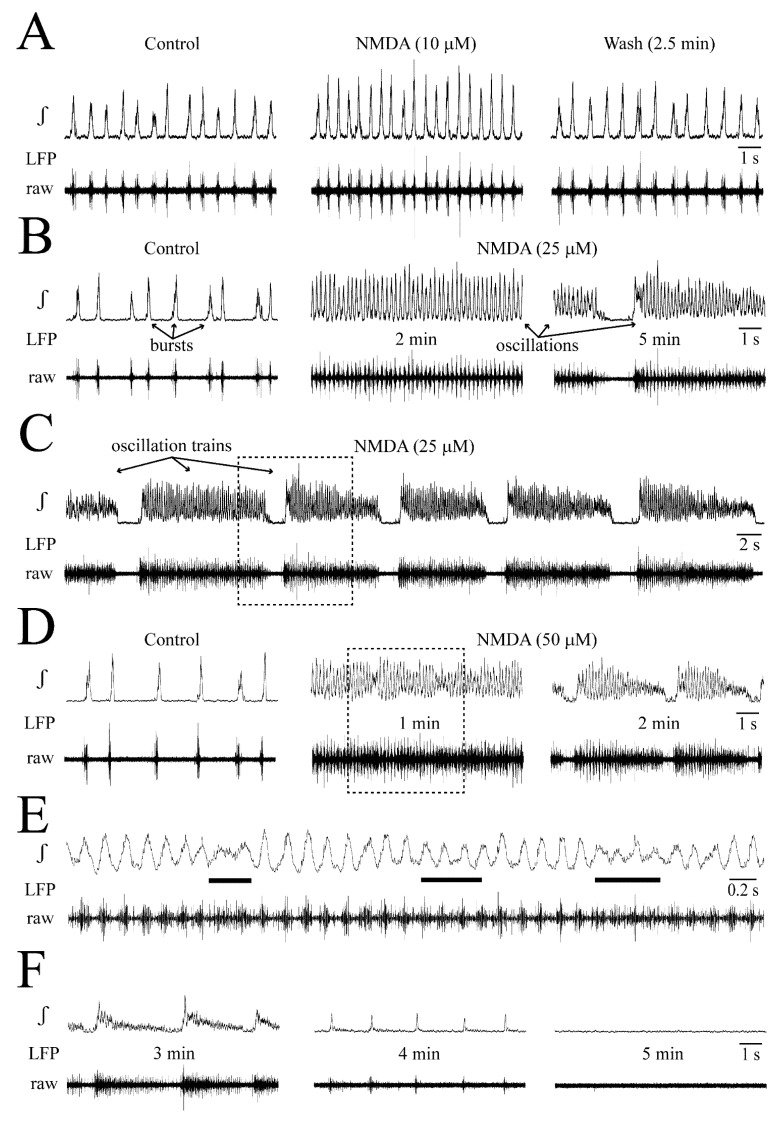
**NMDA effects on LFP.** (**A**) A dose of 10 μM NMDA increased LFP rate and shortened its duration. (**B**) At 2 min, 25 μM NMDA evoked fast ‘oscillations’ with lack of a time period of inactivity between events like in control (compare left and middle panels). At 5 min, oscillations were transiently depressed (right panels). (**C**) Several cycles of resulting ‘oscillation trains’ at lower time resolutions [train in (**B**), indicated by dashed-line box]. Initially, early oscillation and train oscillation rates and durations were similar. However, over several train cycles, oscillation amplitude diminished and rhythm became irregular toward the end of each event. (**D**) At 1 min, 50 μM caused oscillations similar to those in 25 μM, though at this time amplitude fluctuations occurred (middle panels). Dashed-line box indicates the time period for which oscillations are displayed at higher time resolution in (**E**) showing on raw trace smaller amplitude integrated signals are due to tonic activity. Right panels in (**D**) also show augmented decline of amplitude, duration and regularity of oscillation trains at 2 min, compared to effects in 25 μM at ~5 min. (**F**) At 3 min of the experiment in (**C**–**E**), oscillation trains were notably attenuated (left), whereas only a spike-shaped rhythm remained at 4 min and no activity was seen at 5 min.

**Figure 4 brainsci-12-00651-f004:**
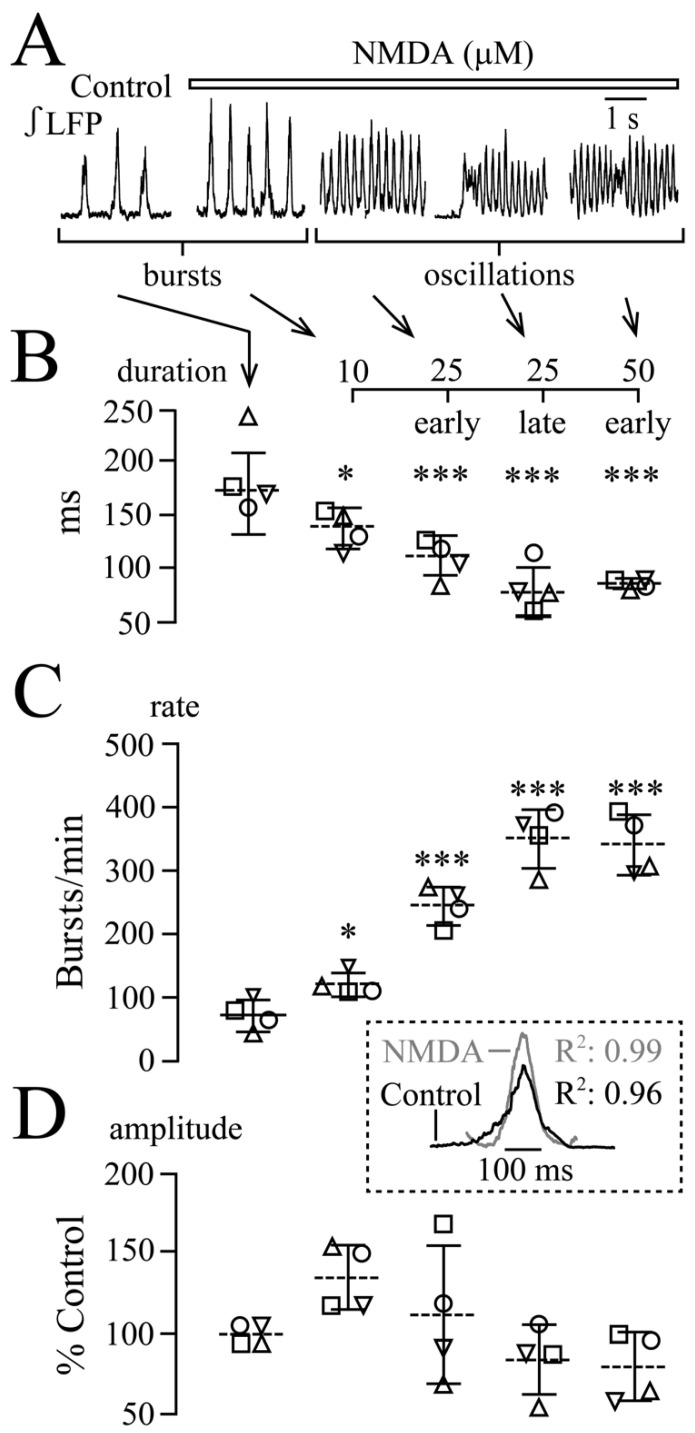
**Quantification of NMDA-evoked LFP pattern transformations exemplified in Figure 3**. (**A**) ‘Early’ LFP oscillations occurred during the first 1–3 min in 25 and 50 μM. In the ‘late’ phase, between 3–5 min (or at the start of wash), oscillation trains developed in 25 and 50 μM. (**B**–**D**) Analysis in four slices indicated by each symbol revealed that single event duration decreased (**B**) and rate (**C**) increased progressively at all doses, whereas amplitude (**D**) did not change. The inset shows for averaged integrated traces from five consecutive events in three of the four slices that the LFP has a Gaussian shape in both control LFP and NMDA. Lines indicate mean values (dotted line) ± SD (solid line), significance was determined with one-way ANOVA with Dunnett’s post test (F (1, 5) = 13.91, *p* < 0.0001, ANOVA). * *p* < 0.05, *** *p* < 0.001).

**Figure 5 brainsci-12-00651-f005:**
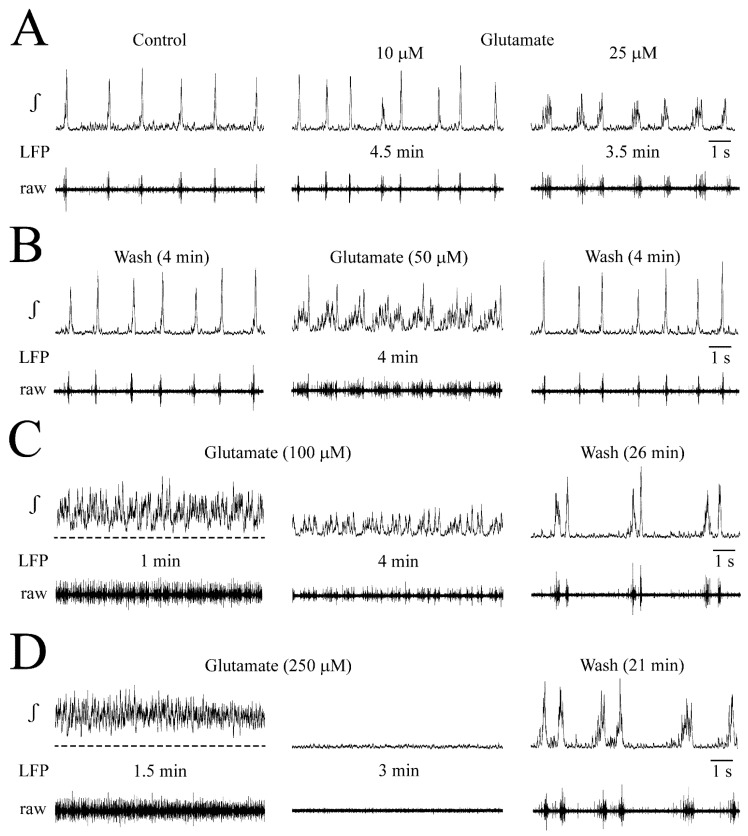
**Glutamate effects on LFP pattern.** Recording is from a single slice. (**A**) A dose of 10 μM glutamate had no effect, whereas 25 μM changed the LFP pattern to longer-lasting, smaller amplitude events. (**B**) After wash, 50 μM evoked almost merged, crescendo-shaped multipeak events. (**C**) After wash, 100 μM at 1 min raised integrated trace baseline as an indication of tonic activity evident on raw trace, but still showed groups of multipeak events before rhythm stopped at 4 min. Recovery resulted in same number of events as in control, but they appeared now as doublets. (**D**) 250 μM caused initially massive tonic activity before rhythm ceased at 3 min. Recovery after 21 min re-established rhythm at a rate similar to control. This time, LFP doublets were intermingled with single events and all had crescendo-like shapes similar to those in 25 and 50 μM glutamate.

**Figure 6 brainsci-12-00651-f006:**
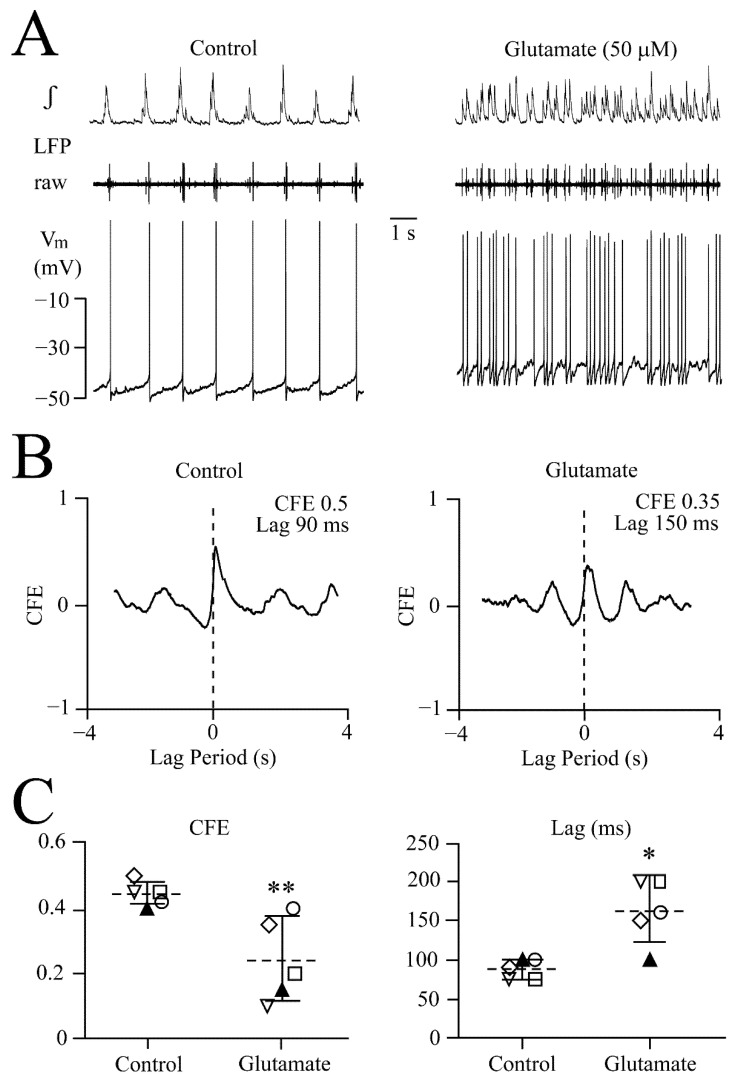
**Cross-correlation of desynchronizing glutamate effects.** (**A**) Left panel shows that a whole-cell-recorded LC neuron ‘spiked’ at the peak of membrane potential (V_m_) ‘subthreshold oscillations’ (STOs) that were in phase with the simultaneously recorded LFP. Right panel shows that 50 μM glutamate, 2 min after start of application, transformed the LFP pattern to irregular discharge accompanied by a neuronal depolarization of 7 mV and accelerated spiking with concomitant ~20% amplitude decrease that was reversible, as evident from the continuous recording from that neuron shown in Figure 10A. (**B**) Cross-correlogram between neuronal spike and LFP during control (left panel) and glutamate (right panel) shows a desynchronizing effect on the LC neuron network, indicated by a decreased correlation function estimate (CFE) value and increased lag time. CFE is the measure of the peak value on the *Y* axis and lag time indicates the shift of peak from 0 time on the *X*-axis. (**C**) Scatter plots from the neuron in (**A**) (filled symbol) and four other neurons reveal a difference in CFE values (left) and lag times (right) for control vs. glutamate and determined with two-tailed paired *t*-test. * *p* = 0.034, ** *p* = 0.0072.

**Figure 7 brainsci-12-00651-f007:**
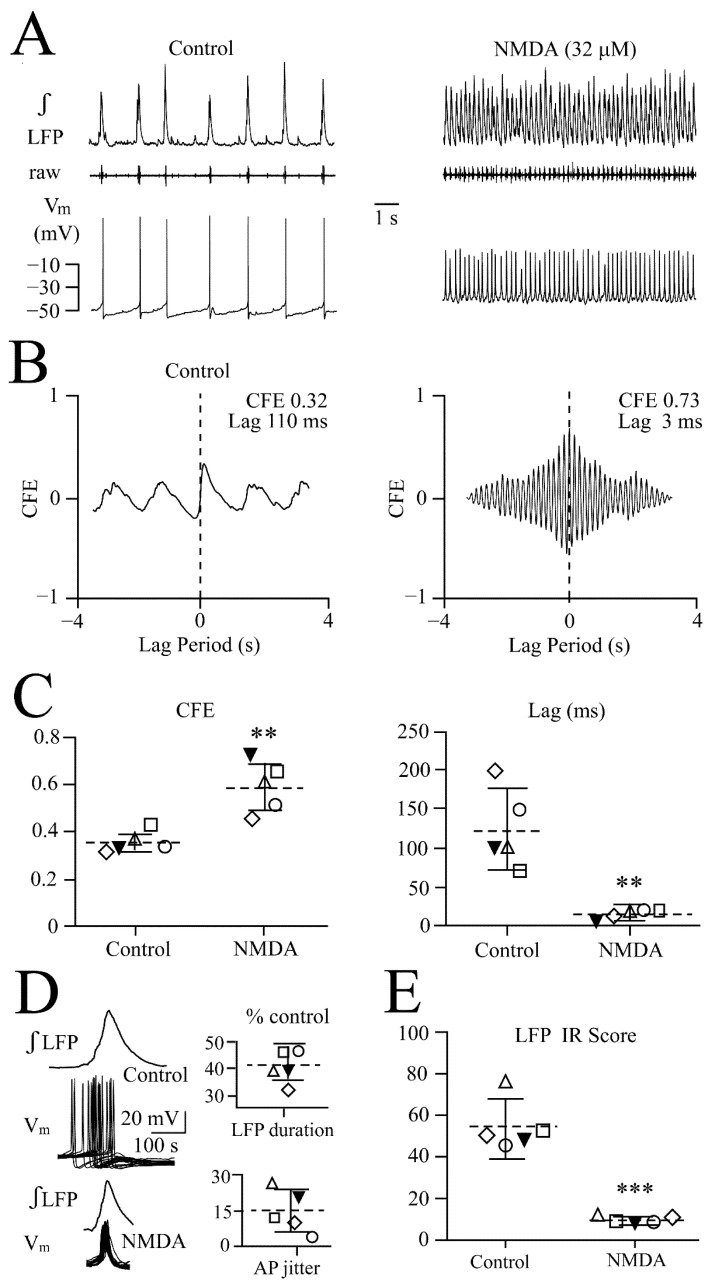
**Cross-correlation of synchronizing NMDA effects.** (**A**) Shows the ‘early’ LFP pattern transformation to faster oscillations in 32 μM NMDA. In the simultaneously recorded neuron, this transformation was accompanied by a V_m_ depolarization of 10 mV and accelerated spiking with a concomitant ~20% amplitude decrease that was reversible, as evident from the continuous recording from that neuron shown in Figure 8A. (**B**) Cross-correlogram between neuronal spike and LFP during control (left panel) and NMDA (right panel) shows a synchronizing effect on the LC neuron network, indicated by a higher CFE value and a lower lag time. (**C**) Scatter plots from the neuron in (**A**) (filled symbol) and four other neurons reveal significant difference in CFE values (left) and lag time (right) for control vs. NMDA determined with the two-tailed paired *t*-test ** *p* = 0.0059 and ** *p* = 0.009, respectively. (**D**) Overlaid traces (also averaged for LFP signal) from 20 consecutive events in control (upper traces) and NMDA (lower traces) in the neuron in (**A**) unravel a notably reduced jitter of cellular spiking in NMDA, plotted as a % change from control in lower right scatter plot for this neuron (filled symbol) and the other four cells. Upper scatter plot represents the % change in integrated LFP duration for the same experiments. (**E**) Scatter plots show for the neurons in (**C**,**D**) that the irregularity (IR) score decreased notably. Significance for plots determined with the two-tailed paired *t*-test (*** *p* < 0.001).

**Figure 8 brainsci-12-00651-f008:**
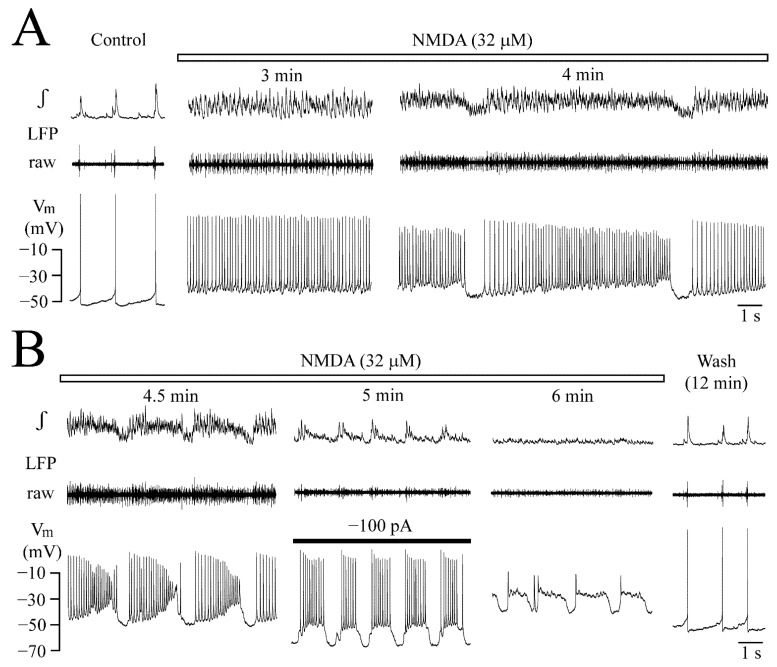
**NMDA effects on neuronal V_m_.** (**A**) In continuation of the recording from the neuron shown in Figure 7A (left and middle panels), NMDA-evoked V_m_ oscillations started to become interrupted by ~1 s-lasting rhythmic hyperpolarizations causing spike blockade. The resulting LFP oscillation trains started after the inactivity period with concomitant progressive V_m_ depolarization leading to accelerated spiking at decreased amplitude (right panels). (**B**) At 4.5 min into NMDA, LFP oscillation trains became shorter in synchrony with V_m_ oscillation time periods (left panels). V_m_ oscillations persisted 30 s later during hyperpolarization to −68 mV due to dc current injection via the patch electrode. 30 s after cessation of current injection, LFP was blocked at 6 min NMDA, but tonic spiking from some neurons was still evident in the raw signal. The neuron showed at a more depolarized level V_m_ hyperpolarizations with sharp offset evoking a small ‘rebound’ spike. Both V_m_ and LFP recovered 12 min after start of wash.

**Figure 9 brainsci-12-00651-f009:**
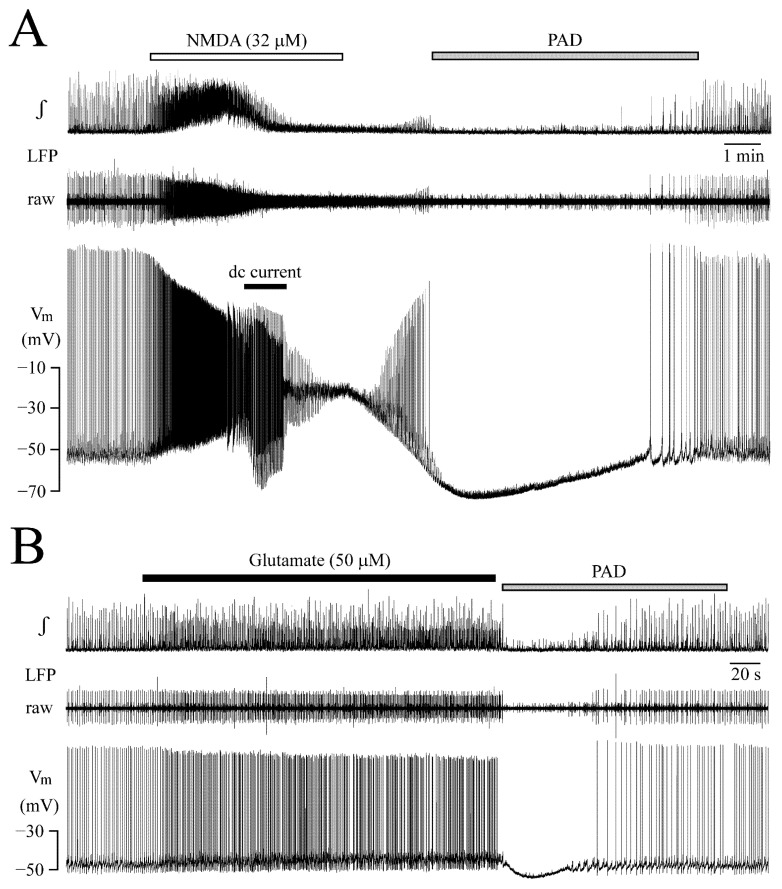
**Post-agonist depression (‘PAD’).** (**A**) Continuous recording from the neuron of Figure 7 and Figure 8 on a condensed time scale. This highlights that immediately upon start of NMDA washout, the neuron starts to hyperpolarize concomitantly with partial recovery of blocked LFP rhythm. However, after 2 min (in synchrony with pronounced neuronal hyperpolarization) cellular spikes and also LFP were blocked again by PAD. Neuronal spiking and LFP rhythm resumed after about 5 min. (**B**) Example for a corresponding glutamate-evoked PAD recorded in a different neuron.

**Figure 10 brainsci-12-00651-f010:**
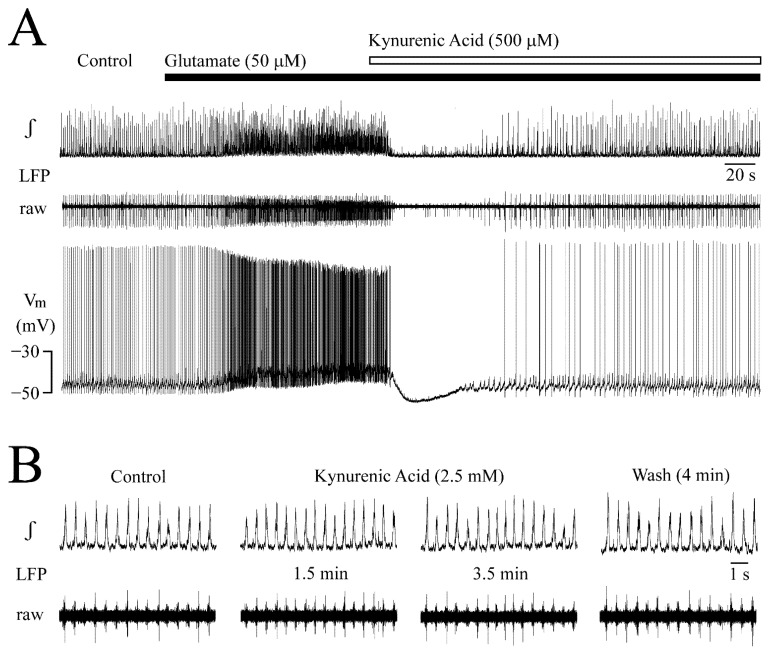
**Receptor-specificity of glutamate effects.** (**A**) Shows for the same neuron illustrated in Figure 6 the LFP pattern-transforming response to glutamate on a condensed time scale. In the initial phase of application of the non-selective ionotropic glutamate receptor blocker kynurenic acid (500 μM), LFP and neuronal spiking were abolished along with a pronounced hyperpolarization. Subsequently, LFP, V_m_ and intracellular spiking recovered to control levels during continuous application of kynurenic acid in glutamate. (**B**) The initial inhibitory kynurenic acid action on LFP and neuron discharge depended on the presence of glutamate as an addition of the drug to control superfusate had no effect on LFP, as measured in a different slice.

## Data Availability

Not applicable.

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
