# Peer review of "NMDA Enhances and Glutamate Attenuates Synchrony of Spontaneous Phase-Locked Locus Coeruleus Network Rhythm in Newborn Rat Brain Slices"

_brainsci, 2022, doi:10.3390/brainsci12050651_

Round 1
Reviewer 1 Report
See uploaded file (Rawal et al)

Reviewer 2 Report
Comments
The authors of the manuscript entitled: "NMDA enhances and glutamate attenuates synchrony of spontaneous phase-locked locus coeruleus network bursting in newborn rat brain slices" described changes in synchrony of spontaneous phase-locked locus coeruleus bursting by NMDA and glutamate. To publish this manuscript in Brain Sciences, the authors should try to do the following suggestions:
-This MS, while valuable and generally well-written, contains many grammatical errors and I suggest that the MS be edited by an English-language specialist.
-Abstract needs to be revised and rewritten.
Material and methods
The number of animals used in some experiments is not clear.
Results
-The first sentence is long and incomprehensible.
-Instead of LC it should be better to use LC neurons.
-LFP burst rate, duration, and amplitude were regular. This sentence is better to rewrite: Burst rate, duration, and amplitude of LFP Were regular
-The typically sparse irregularity of LFP bursts between ……is incomprehensible.
-Why Figure 1A-D is not in time order? There is no explaining what are the mentioned times?
-The title of figure 2: LC neuron spike jitter and failure are not grammatically correct.
-In the legend of figure 3 what is the phase like. It seems that it is phase lock.
-‘’LFP duration (in ms) decreased from 183 ± 38 to 133 ± 18’’ is better to be replaced by ‘’LFP duration decreased from 183 ± 38 to 133 ± 18 ms’’.
-In figure 8, it is better to add the graph.
-In figure 10, it is better to add the graph.
Discussion
-The first sentence is long and incomprehensible.
The discussion needs more citations regarding LC recording
Reviewer 3 Report
Manuscript ID: brainsci-1679001
In the present manuscript, the authors evaluate the effect of NMDA and glutamate on electrophysiological properties of locus coeruleus in newborn rat. Interestingly, they found a differential effect on spontaneous phase-locked locus coeruleus network bursting of neonatal rats. However, authors should be cautious about jumping to conclusions about results obtained. In my opinion, results are difficult to interpret. Figures and legends should have more detailed information. This information will improve the interpretation and comprehension of the results, as well as, to discussion of results.
Round 2
Reviewer 1 Report
No further comment